

# Atrial fibrillation and psychological factors: a systematic review

Federica Galli[1], Lidia Borghi[1], Stefano Carugo[1,2], Marco Cavicchioli[3], Elena Maria Faioni[1,4], Maria Silvia Negroni[2] and Elena Vegni[1]

[1] Department of Health Sciences, University of Milan, Milan, Italy
[2] Cardiology Unit and UTIC, UOC Cardiology, ASST Santi Paolo e Carlo, Milan, Italy
[3] Vita-Salute San Raffaele University, Milan, Italy
[4] SIMT, ASST Santi Paolo e Carlo, Milan, Italy

Corresponding author
Federica Galli,
federica.galli1@unimi.it

## ABSTRACT

**Background.** Psychological factors have been suggested to have an influence in Atrial Fibrillation (AF) onset, progression, severity and outcomes, but their role is unclear and mainly focused on anxiety and depression.

**Methods.** A systematic electronic search had been conducted to identify studies exploring different psychological factors in AF. The search retrieved 832 articles that were reviewed according to inclusion criteria: observational study with a control/comparison group; use of standardized and validated instruments for psychological assessment. Results were summarized qualitatively and quantitatively by effect size measure (Cohen's d and its 95% confidence interval). Cochrane Collaboration guidelines and the PRISMA Statement were adopted.

**Results.** Eight studies were included in the systematic review. Depression was the most studied construct/ but only one study showed a clear link with AF. The remaining studies showed small and non-significant (95% CI [−0.25–1.00]) differences between AF and controls, no differences in frequency of depression history (95% CI [−0.14–0.22]) or in case frequency (95% CI [−0.50–0.04]). Miscellaneous results were found as far as anxiety: AF patients showed higher levels when compared to healthy subjects (95% CI [2.05–2.95]), but findings were inconsistent when compared to other heart diseases. Considering personality and life-events preceding AF, we respectively found a large (95% CI [1.87–2.49]) and a moderate to large effect (95% CI [0.48–0.98]).

**Discussion.** The small number of studies does not allow to draw clear-cut conclusions on the involvement of psychological factors in AF. Promising lines of research are related to personality and adverse life-events, and to the increase of longitudinal design studies. Some methodological problems could be overcome by including clinical psychologists in the implementation of research protocols.

## INTRODUCTION

Atrial fibrillation (AF) is the most common arrhythmia in clinical practice, with an overall prevalence of 1–2% in the general population (*Lip, Tse & Lane, 2012*; *Piccini et al., 2012*) and an incidence that increases with age. In Europe, 3.7–4.2% in the age range of 60–70

and 10–17% of those 80 years or older are affected by AF, and, in the next 50 years, its prevalence is expected to double, as a consequence of the prolongation of life expectancy (*Zoni-Berisso et al., 2014*).

On the basis of the presentation, duration, and spontaneous termination of AF episodes, five types of AF are classified: first diagnosed, paroxysmal, persistent, long-standing persistent, and permanent AF (*Kirchhof et al., 2016*). AF is associated with a higher relative risk of all-cause mortality, stroke, cardiovascular mortality, cardiac events, and heart failure (more in women than in men) (*Emdin et al., 2016*) and a number of risk factors (see Table A.1). However, to the best of our knowledge, the studies on risk factors did not explore the role of psychological factors. The lack of correspondence between symptoms and ECG findings (*Lampert, 2015*), may be clarified evidencing if and which psychological factors are linked to AF. Two reviews analysed the link of AF with anxiety and depression (*McCabe, 2010*; *Patel et al., 2013*) suggesting a condition of comorbidity. However, the comorbidity of anxiety and depression is common in so many and different disorders and diseases (*American Psychiatric Association, 2013*; *Galli et al., 2017*; *Galli, 2017*) that we need to enlarge our perspective. Temperament, adverse childhood experience, stressful life events and personality are well known psychological mechanisms representing risk and prognostic factors for anxiety and mood disorders, (*American Psychiatric Association, 2013*). In this perspective, the aim of the present study is to make a systematic review of the numerous studies dealing with psychological factors in AF, in order to identify factors not simply linked to AF, but also influencing its onset, severity and clinical outcome. To the best of our knowledge, no systematic reviews have been realised on the role of psychological factors in AF. The recognition of the involvement of such factors may help the identification of new clinical strategies for the management of AF.

## METHODS

### Search strategy

To include the broadest range of relevant literature, electronic searches were conducted on the major databases in the field of health and social sciences: Pubmed, Scopus, Embase, PsycInfo, and Web of Science. The search was performed using Mesh terms/Keywords (depending on the database) with the same search strategy: "Atrial fibrillation" AND "Psychological distress" OR "Anxiety" OR "Depression" OR "Emotional distress" OR "Personality" OR "Psychiatric disorder" OR "Temperament" OR "Life-event". The selection of the search terms are based on the clinical experience and the topic literature on psychological factors involved in physical disorders (*American Psychiatric Association, 2013*). The search was limited to English-written publications, and to the period from 2000 to the present, to focus on contemporary psychological models, theories, and assessment tools. An additional analysis of the reference list in each selected paper was also performed. When the full text was not retrievable, the study was excluded.

The electronic bibliographic search was conducted in December 2016.

## Selection criteria and data extraction

Inclusion criteria:

- Studies with an analytical study design as defined by *Grimes & Schulz (2002)* (i.e., an observational study with a comparison or control group);
- Studies involving patients with a diagnosis of AF (*Fuster et al., 2011*);
- Studies adopting standardised and validated instruments to assess psychological factors;
- Studies written in English language.

  Exclusion criteria:

- Case reports, reviews, Letters to the Editor, meeting abstracts, book chapters;
- Pharmacological and behavioural intervention trials, surgical protocols, or validation of measurement instruments;
- Studies with intra-group controls (e.g., men *vs* women; stratification according to AF severity or AF type);
- Studies addressing only quality of life.

## Data extraction

Study selection was performed by two independent reviewers with research expertise in clinical psychology (FG and LB) who assessed the relevance of the study for the objectives of this review. This first round of selection was based on the title, abstract, and keywords of each study. If the reviewers did not reach a consensus or the abstract did not contain sufficient information, the full text was reviewed.

In the second phase, full-text reports have been evaluated to detect whether the studies met the inclusion criteria (Fig. 1).

A standardised data extraction form was prepared; data was independently extracted by two of the authors (FG and LB) and inserted in a study database. A process of discussion/consensus moderated by a third reviewer (EV) (*Furlan et al., 2009*) resolved discrepancies between reviewers.

## Statistical methods

A systematic analysis was conducted according to the Cochrane Collaboration guidelines (*Higgins & Green, 2011*) and the PRISMA Statement (*Moher et al., 2009*). As the included studies were highly heterogeneous in terms of participants, variables, instruments, and outcomes, it was not considered appropriate to undertake a meta-analysis (*Higgins & Green, 2011*); however, effect size computations were performed using Cohen's *d* (*Cohen, 1988*) and its 95% confidence interval (*Borestein et al., 2011*) for each outcome measure within each study. The index was primarily calculated using descriptive statistics reported in the results section of each study. When binary data was reported, we estimated the Odds Ratio and then we applied the appropriate procedures (*Borestein et al., 2011*) to convert it to *d.* Cohen's *d* values less than or equal to 0.20, 0.50 and 0.80 were interpreted as small, medium and large effect size respectively (*Cohen, 1988*). Furthermore, in order to assess possible confounding factors, such as sample size and year of publication, that might
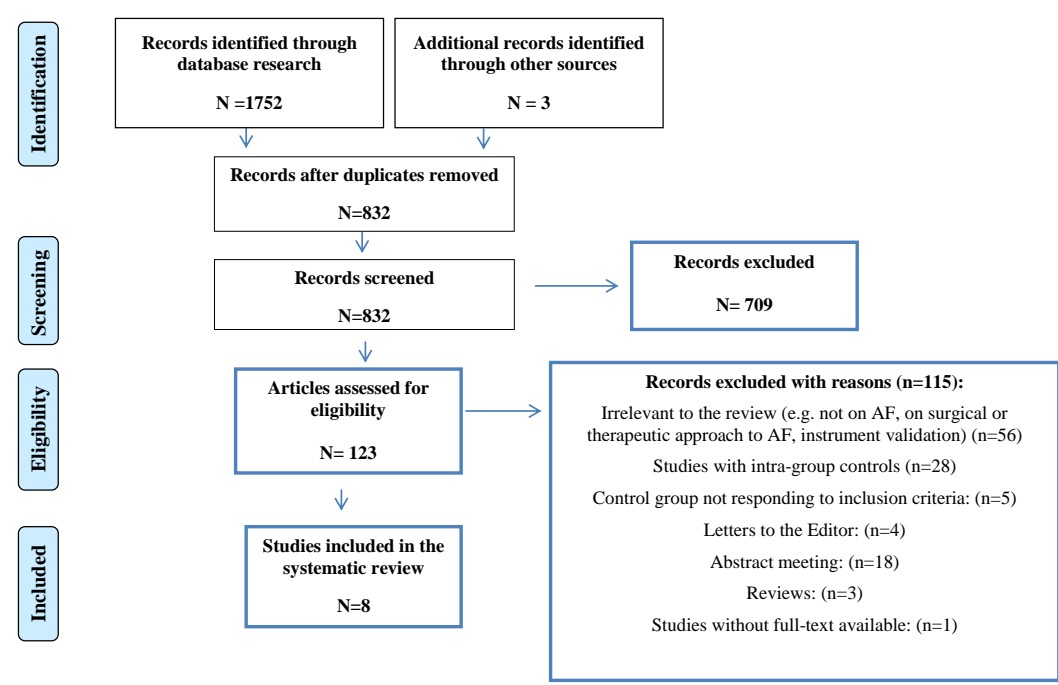

**Figure 1** Flow diagram of literature search and selection of publications.

influence effect size values, we estimated correlations using Spearman's rank correlation based on bootstrap methodology (*bias corrected and accelerated*) (*Davidson & Hinkley, 1997*). 1,000 bootstrap independent samples were used with a *p* (2-tailed) <0.05.

## Risk of bias

In order to assess the risk of bias of the included studies, we adopted a tool for assessing risk-of-bias in nonrandomized studies developed by Kim and colleagues *(2013)*. In brief, two reviewers (MC and FG) independently extracted relevant information and data from all eligible reports, and they independently applied criteria for judging the risk of bias for each domain. Particularly, the Risk of Bias Assessment Tool for Nonrandomized Studies (RoBANS) evaluates six domains with three different levels of risk of bias (i.e., high, low, unclear): (a) selection of participants; (b) confounding variables; (c) measurement of exposure; (d) blinding of outcome assessments; (e) incomplete outcome data; (f) selective outcome reporting.

## RESULTS

We found eight studies that adopted a control group (Fig. 1), for a total of 98,5641 subjects (89,383 AF patients; 896,109 healthy controls; 149 other heart diseases). The psychological variables detected by the selected studies are described in Table 1. Table 2 summarizes the main characteristics and outcomes of the included studies.

Five studies (*Thrall et al., 2007*; *Dkabrowski et al., 2010*; *Perret-Guillaume et al., 2010*; *Schnabel et al., 2013*; *Lioni et al., 2014*) evaluated the role of depression in AF using self-report instruments. Nevertheless, we observed a high heterogeneity in the construct assessed

Galli et al. (2017), *PeerJ*, DOI 10.7717/peerj.3537
## Table 1 Overview of the selected studies.

| First Author, Year | N Sample (mean age, SD) | N Controls (mean age, SD) | AF diagnosis[a] | AF pattern | Questionnaire (s) | Study design | Significant findings compared to control(s)/(p) | Effect size d (95% CI) | Note |
|---|---|---|---|---|---|---|---|---|---|
| Mattioli et al. (2005) | 116 (54 ± 7) | 116 (54 ± 6.5) | - "new and clearly recognizable onset of symptoms including palpitations, dyspnea or dizziness, or a combination of these symptoms". Confirmation by ECG. | - First diagnosed acute episode of lone AF | - Minnesota Multiphasic Personality Inventory (MMPI-2)-Type A scale. - Life Changes Scaling. | Cross-sectional | - pattern of Type A behavior in 20% of AF vs 9% of controls ($p < .001$) -Life Changes Unit in AF was 56 ± 33 vs 34 ± 27 in controls ($p < .01$). | d pattern Type behavior 2.18 (1.87–2.49) d life change units 0.73 (0.48–0.98) | -increasing level of Life Changes Units was associated with greater risk of AF. -Spontaneous conversion of AF has predicted by acute stress and type-A behavior. |
| Thrall et al. (2007) | 101 (66.3 ± 11) | 97 (hypertensive) (68 ± 7.2) | - Current criteria | - Recurrent AF (N=59) - Permanent AF (N = 42) | -Beck Depression Inventory (BDI) -State-trait Anxiety Inventory (STAI) | Cross-sectional | - BDI score did not differ from controls both baseline ($p = .32$) and 6-month follow-up ($p = 0.48$) - Trait anxiety differed from controls at baseline ($p = .03$), but not 6-month follow-up (p=.25) | d depression 0.26 (−0.01 −0.53) d anxiety trait .34 (0.07–0.61) | - no influence of depression on AF - Higher level of trait anxiety in AF (only baseline), but STAI anxiety is a measure of a stable psychologically trait, and changes in the little period is enigmatic. -no data have been reported for state anxiety |
| Dkabrowski et al. (2010) | 150 (men: 67.8 ± 10.5); (women: 64.1 ± 9.5) | 70 (men:56.5 ± 13.3); (women: 54.8 ± 12.5) | -By exclusion: AF due to valvular heart disease, with valve prosthesis, cardiomyopathy, heart failure or left ventricular dysfunction (ejection fraction <55%) were excluded. | - Paroximal AF (N = 61) -Persistent AF (N = 46) -Permanent AF (N = 43) | -Beck Depression Inventory (BDI) | Cross-sectional | - BDI scores showed more severe symptoms indicating depression in all subsets of patients with AF compared to controls ($p < .01$) - Higher level of depression in women ($p < .0005$) than men (only in AF group) | d overall depression 0.38 (−0.25–1.00) | -All forms of AF have substantial impact on the risk of depression occurrence. |
| Perret-Guillaume et al. (2010) | 41 (72.3 ± 3.9) | 123 (72 ± 4) | -Permanent AF in 30 cases, paroxysmal AF in 7 and new diagnosis in 4. | - Not specified | - Duke Health Profile (Duke) - SF-36 | Cross-sectional | -Anxiety ($p = .03$) and Depression ($p = .003$) statistically relevant in AF compared to controls. - DUKE: mental score more impaired in AF than controls ($p < .01$) - SF-36: Mental Health score does not differ from controls ($p = .61$) | d anxiety 2.50 (2.05–2.95) d depression 3.08 (2.63–3.57) d mentalhealth 0.55 (0.20–.90) | - Mental, Anxiety and Depression dimensions remained impaired even when adjusted for potential clinical confounding factors (coronary artery disease or chronic respiratory failure). - Mental Health scores disagree comparing the two different assessment tools. |

Galli et al. (2017), *PeerJ*, DOI 10.7717/peerj.3537

Galli et al. (2017), *PeerJ*, DOI 10.7717/peerj.3537

**Table 1** (*continued*)

| First Author, Year | N Sample (mean age, SD) | N Controls (mean age, SD) | AF diagnosis[a] | AF pattern | Questionnaire (s) | Study design | Significant findings compared to control(s)/(p) | Effect size d (95% CI) | Note |
|---|---|---|---|---|---|---|---|---|---|
| *Whang et al. (2012)* | 771 | 730 | "An endpoint committee of physicians reviewed medical records for reported events according to predefined criteria. An incident AF event was confirmed if there was electrocardiographic evidence for AF…" | Not specified | Mental Health Inventory-5 | Longitudinal | Comparison of AF with no-AF group with the least global distress score (NS) | *d psychological distress* 0.12 (−0.28 - 0.04)) | The psychological distress did and a proxy measure of depression did not differ between the groups. |
| *Schnabel et al. (2013)* | 309 (64.8 ± 8.2) | 9.680 (55.2 ± 10.8) | -History of self-reported AF and/or electrocardiographic documentation | - Not specified | - Patient Health Questionnaire (PHQ-9) - Computer-assisted question on the previous history of any depressive disorder as diagnosed by a physician. | Cross-sectional | - no cases of depression (PHQ-9 ≥ 10): 7.3% of controls *vs* 5.8% of AF - History of depression:16.2% of AF *vs* 15.4% of controls - more pronounced somatic symptom dimension of depression in multivariate logistic regression | *d history of depression* 0.04 (−0.14–0.22) *d caseness of depression* −0.23 (−0.50–0.04) | - Age range: 35–74 years - Unclear as "mental health status" has been assessed - the conclusion of "higher burden of depressive symptoms" is not supported by the analysis of reported data |
| *Lioni et al. (2014)* | 54 (56.64 ± 12.50) | 52 (40.46 + 14.96 (supraventricolartachycardias (SVTs)) | - "current guidelines" -patients were referred for catheter ablation (symptomatic, drug-refractory, paroxysmal AF vs symptomatic SVTs) | - Paroxysmal AF | -Beck Depression Inventory (BDI) -State-trait Anxiety Inventory (STAI) | Cross-sectional | - STAI state differed from controls (*p* < .01) - STAI trait slightly differed from controls (*p* < .05) - BDI did not differ from controls (*p* = .077) | *d anxiety trait* 0.41 (0.02–0.80) *d anxiety state* 0.77 (0.38–1.16) *d depression* 0.35 (0.02–0.72) | - patients "referred for catheter ablation" may not be representative of the AF population (usually older as well). |
| *Graff et al. (2016)* | 88.612 (NA) | 886.120 (NA) | -first-time inpatient diagnosis of AF by ICD-8 and ICD-10 | - First diagnosed AF | - Danish Civil Registration System (for identifying spousal/partner death). | Longitudinal | -partner bereavement was experienced by 144 AF and 1036 controls. - Transient higher risk (41%) of developing AF within 30 days after death in the bereaved population. | *d bareavement and AF* 0.19 (−0.14–0.52) | - The risk of AF lasts about one year and it is especially high for those who were young and those who lost a relatively young partner. |

**Notes.**

NA, Not Addressed.

[a] AF diagnosis is reported referring to what declared by the authors of each study.

Table 2  Risk of bias.

| Studies | Selection of participants | Confounding variables | Measurement of exposure | Blinding of outcome assessments | Incomplete outcome data | Selective outcome reporting |
|---|---|---|---|---|---|---|
| Mattioli et al. (2005) | Green | Green | Red | Red | Green | Green |
| Thrall et al. (2007) | Green | Green | Red | Red | Green | Red |
| Dkabrowski et al. (2010) | Green | Red | Red | Red | Green | Green |
| Perret-Guillaume et al. (2010) | Green | Green | Red | Red | Green | Green |
| Whang et al. (2012) | Green | Green | Red | Red | Green | Green |
| Schnabel et al. (2013) | Green | Green | Red | Red | Green | Green |
| Lioni et al. (2014) | Green | Red | Red | Red | Green | Green |
| Graff et al. (2016) | Green | Green | Green | Green | Green | Green |

Notes.
Red, High risk; Green, Low risk; Yellow, Unclear.

and in the questionnaires adopted in each study. Three studies assessed anxiety, both in term of level of state (current) and trait (lifetime) (*Thrall et al., 2007*; *Perret-Guillaume et al., 2010*; *Lioni et al., 2014*)

Two studies investigated the role of life events on AF onset: one study highlighted the role of bereavement (*Graff et al., 2016*) and another the influence of Life Changes Unit (*Mattioli et al., 2005*).

One study evaluated the role of personality traits (*Mattioli et al., 2005*); while two studies (*Perret-Guillaume et al., 2010*; *Whang et al., 2012*) explored general mental health (*Perret-Guillaume et al., 2010*) and global psychological distress (*Whang et al., 2012*) with contrasting findings.

Considering depression, only the study of *Perret-Guillaume et al. (2010)* found a large significant (95% CI [2.63–3.57]) difference in level of depression between AF patients and healthy controls. The remaining studies showed a small and non-significant (95% CI [−0.14–0.52]) difference between individuals affected by AF and healthy controls. Considering the comparison between AF patients and subjects with other heart diseases, we found small and null effect sizes for both studies included in the current work (*Thrall et al., 2007*; *Lioni et al., 2014*) (respectively: $d = 0.35\,[0.02 - 0.77]$; $d = 0.26[-0.01 - 0.56]$).

Miscellaneous results were found concerning the role of anxiety. Particularly, AF subjects exhibited more severe anxiety symptoms than healthy subjects (*Schnabel et al., 2013*) (95% CI [2.05–2.95]). Inconsistent findings, instead, were present when anxiety (STAI state) was compared between AF and other heart diseases. Specifically, Thrall and colleagues (*2007*) found no significant differences in STAI state scores between AF and hypertensive patients. Conversely, Lioni and colleagues (*2014*) showed that AF patients were characterized by a more intense anxiety state than controls ($d = 0.77\,[0.38 - 1.16]$). On the other hand, consistent results were found when levels of anxiety trait were evaluated comparing AF to other heart diseases: AF showed higher levels of anxiety trait than Supraventricular Tachycardias subjects (*Lioni et al., 2014*) and hypertensive patients (*Thrall et al., 2007*). In detail, this difference was small to moderate and significant in both studies (*Thrall et al., 2007*; *Lioni et al., 2014*) (respectively: $d = 0.41\,[0.02–0.80]$; $d = 0.34\,[0.07–0.61]$).

Considering the role of personality, a significant large (95% CI [1.87–2.49]) difference was found. Regarding the role of life events, the two studies (*Mattioli et al., 2005*; *Graff et al., 2016*) reported inconsistent results: the study on the Life Changes Units preceding AF (*Mattioli et al., 2005*) reported a moderate to large (95% CI [0.48–0.98]) difference, while the study on partner bereavement reported a small and insignificant effect ($d = 0.19$ [−0.14–0.52]).

One study evaluated general mental health using the DUKE test (*Schnabel et al., 2013*), showing a moderate and significant (95% CI [0.20–0.90]) effect size; conversely, the mental health score evaluated by the SF-36 did not differ significantly ($p = 0.61$). The longitudinal investigation (*Whang et al., 2012*) on the role of the global psychological distress did not evidence any significant difference for AF patients ($d = −.22$ [−.44–.00]).

We then checked the effect of possible confounding factors such as sample size and year of publication, we did not find any significant relations with effect size values. Consequently, we might exclude that these aspects affected the extent of difference between AF patients and controls. The risk of bias is described in Table 2.

## DISCUSSION

The role of psychological factors has been extensively studied in AF, but any clear-cut conclusion is far to be reached on the basis of our systematic review. Principally, we have to stress the paucity of studies that we could select.

Our review shows that depression and anxiety remain the most studied psychological features in AF. Findings from longitudinal studies do not allow any conclusion on the direction of the association. Findings on depression (*Thrall et al., 2007*; *Dkabrowski et al., 2010*; *Perret-Guillaume et al., 2010*; *Schnabel et al., 2013*; *Lioni et al., 2014*) are difficult to compare and conclusions are far from consistent. Two unstructured reviews on anxiety and depression (*McCabe, 2010*; *Patel et al., 2013*), suggested a generic condition of comorbidity. In our review, only two studies supported a role for depression (*Dkabrowski et al., 2010*; *Perret-Guillaume et al., 2010*). However, in one study (*Perret-Guillaume et al., 2010*) the assessment tool (Duke Health Profile) was aimed at detecting the symptoms of depression (and anxiety) in the week before the interview, so that any conclusion pertains only to that time span. *Dkabrowski et al. (2010)* found a higher prevalence of depression level in AF patients (especially in women) than controls, but the effect size does not allow to support any rigorous conclusion.

Three studies assessed anxiety (*Thrall et al., 2007*; *Perret-Guillaume et al., 2010*; *Lioni et al., 2014*) and conclusions, similarly to depression, are difficult to draw. We have the same comments and conclusion reported on depression for the study of *Perret-Guillaume et al. (2010)*. In the two other studies (*Thrall et al., 2007*; *Lioni et al., 2014*) trait anxiety slightly differed in AF patients compared to controls, but such a difference inexplicably disappeared after six months (*Thrall et al., 2007*). In fact, trait anxiety refers to a stable lifelong component of the personality, that is unlikely to change spontaneously in the brief period.

In synthesis, we can state that the role of anxiety and depression in AF is still largely unknown. Unfortunately, the most studies we reviewed are observational, which are limited in their capacity to infer causality. The little we know on depression probably goes in the direction of being a consequence of permanent AF, and not a triggering factor. The overdrive of the sympathetic nervous system is involved in AF (*Shen & Zipes, 2014*; *Uradu et al., 2017*), and this is the mechanism that might be linked to the role of anxiety in influencing AF. Longitudinal studies are further warranted on this topic, because at the current state of the art we do not know if depressed/anxious patients are more likely to develop AF, or if having AF makes it more likely to develop depression/anxiety. Furthermore, we do not know if a highly symptomatic patients might be more likely to experience anxiety and depression than one who is asymptomatic. It is subsumed that lightening these points may have positive repercussions on the clinical management of these kind of patients.

In the implementation of research protocol, we suggest to differentiate between patients with first diagnosed, paroxysmal, persistent or long-standing persistent, AF (some studies we analysed did not report any specificity on the diagnosis of AF), because the psychological *milieu* may differ accordingly. For a patient is very different to manage a new clinical condition or a long-standing persistent illness. In a similar way, the influences of psychological factors may reverberate differently in a chronic or acute condition, and vice-versa (*Borghi, Galli & Vegni, 2016*). The age of patients is another important variable, because we know that older the patient higher the risk of incidence of AF (*Dewland et al., 2013*), and age needs to be taken into consideration in the implementation of psychological research as well. For the psychological assessment, we suggest the choice of tools based on a life-time period to validate an elderly population (e.g., Geriatric Depression Scale) (*Yesavage et al., 1982–1983*). Furthermore, psychological tests should be administrated in the appropriate context, as psychological assessment may be invalidated by invasive manoeuvres, such as catheter ablation (*Lioni et al., 2014*) or hospitalization (*Perret-Guillaume et al., 2010*). The sole study dealing with personality (*Mattioli et al., 2005*) shows interesting and significant findings for Type-A scale of Minnesota Multiphasic Personality Inventory (MMPI-2), that described AF patients as hard-driving, fast-moving and work-oriented individuals who frequently became impatient, irritable and annoyed. In general, we point out the paucity of studies (also considering not eligible ones) investigating the personality characteristics of AF patients. We think that other promising fields of research could be the study of specific personality traits. Literature shows that personality traits are highly associated with cardiovascular events (*Kubzansky et al., 2006*; *Chida & Steptoe, 2009*), and it can be hypothesized that they also play a role in the development of AF. Allostatic overload is another topic that could be of interest. It describes the cost, in terms of biological burden, of the organism continual adjustment to different challenges (*Fava et al., 2010*). The role of allostatic overload in AF has already been described (*Offidani et al., 2013*), but further studies are required.

The role of adverse life-events is another interesting topic to enhance our knowledge of AF. We included two studies (*Mattioli et al., 2005*; *Graff et al., 2016*) that brought evidence of a role of life-events in predicting the onset of AF, giving even a time span of the increased risk (30 days before the onset of AF). Graff and colleagues (*2016*) characterized life-events, as related to the death of a partner, even if the effect size was small and non-significant. Furthermore, we need to mention the shortcoming of catching data by an administrative database. A population study (*Karatzias, Yan & Jowett, 2015*) showed a significant relationship between the death of a partner and heart disease, but a specific association with AF needs further research. In detail, results reported in these studies might reflect biased effects, particularly as related to assessment procedures.

Consequently, future research should implement study designs with valid and reliable structured clinical interviews in order to adequately evaluate specific psychological conditions.

Our study is not devoid of limitations. We could not perform a meta-analysis. Several studies adopt the Hospital Anxiety and Depression Scale (HADS) (*Zigmond & Snaith, 1983*), but we could not select any of these studies for the inclusion in our systematic review, primarily for the absence of a control/comparison group. HADS is a screening test for detecting symptoms of anxiety and depression in the week before hospitalisation and a valid tool in the older population as well (*Flint & Rifat, 2002*). HADS has been largely adopted to study AF patients (*Gehi et al., 2012*; *Akintade et al., 2015*) and conclusions should always take into consideration that the instrument does not allow to formulate a diagnosis of "anxiety" and/or "depression", and only indicates the presence of referred symptoms.

In synthesis, our review does not allow any straightforward conclusion regarding the role of psychological factors in AF. First, the study of depression seems to indicate it could be a reaction to the diagnosis of AF. To confirm this, we need further studies that include a wider time span, with psychological tests detecting both the current and lifetime situation. The only longitudinal study (*Whang et al., 2012*) did not evidence any relationship, so that we are not allowed to take any conclusion on the likely role of psychological distress as influencing AF. Secondly, the study of personality characteristics is at its beginning in AF, but the knowledge of personality features that may predispose to AF may be useful in a prevention perspective. In general, we stress the need of studies adopting tests suitable for older patients. Finally, the role of life events deserves further attention because evidence in this direction exists, but needs further studies to clarify the specificity of the association.

Finally, we warmly suggest the involvement of clinical psychologists in planning and realizing research in the medical field because the number of suitable instruments to detect the psychological characteristics of patients is very wide, changes with age, and depends on the psychological constructs one chooses to analyse.

# APPENDIX

| Table A.1 | Risk factors associated with the development and progression of atrial fibrillation. |
| --- | --- |
| **Risk factors** | **References** |
| Congestive heart failure | *Benjamin et al. (1994)* |
| Hypertension | *Krittayaphong et al. (2016)* |
| Diabetes mellitus | *Pallisgaard et al. (2016)* |
| Obesity | *Tedrow et al. (2010)*, *Karasoy et al. (2013)* |
| Obstructive sleep apnoea syndrome | *Kanagala et al. (2003)* |
| Chronic kidney disease | *Watanabe et al. (2009)* |
| Liver cirrhosis | *Lee et al. (2017)* |
| Hyperthyroidism | *Woeber (1992)* |
| Genetic factors | *Gutierrez & Chung (2016)* |
| Age | *Dewland et al. (2013)* |
| Smoking | *Zhu et al. (2016)* |
| Alcohol consumption | *Kodama et al. (2011)* |
| Endurance exercise | *Mont, Elosua & Brugada (2009)*, *Myrstad et al. (2014)* |

## Funding

The authors received no funding for this work.

## Competing Interests

The authors declare there are no competing interests.

## Author Contributions

- Federica Galli conceived and designed the experiments, performed the experiments, analyzed the data, wrote the paper, prepared figures and/or tables, reviewed drafts of the paper.
- Lidia Borghi analyzed the data, wrote the paper, prepared figures and/or tables, reviewed drafts of the paper.
- Stefano Carugo conceived and designed the experiments, analyzed the data, wrote the paper, reviewed drafts of the paper.
- Marco Cavicchioli performed the experiments, analyzed the data, prepared figures and/or tables.
- Elena Maria Faioni analyzed the data, reviewed drafts of the paper.
- Maria Silvia Negroni prepared figures and/or tables, reviewed drafts of the paper, last check of the bibliography by the side of cardiological issues.
- Elena Vegni conceived and designed the experiments, analyzed the data, reviewed drafts of the paper.

## Data Availability

The raw data has been supplied as a Supplementary File.

## Supplemental Information

Supplemental information for this article can be found online at http://dx.doi.org/10.7717/peerj.3537#supplemental-information.

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
