# Peer review of "Atrial fibrillation and psychological factors: a systematic review"

_PeerJ, doi:10.7717/peerj.3537_

## Round 0.1 · original submission · Major Revisions

Dear authors,

Three reviewers have analyzed your paper in order to determine if it has high standards to be published in PeerJ. All of them have indicated that your research is scientifically valid once some issues are solved by you. Therefore, my decision is TO REVISE.

With respect and warm regards,
Dr Palazón-Bru (academic editor for PeerJ)

Reviewer 1 ·

Basic reporting

I miss a division of subheadings in the abstract for readability.

Moreover, the risk factors listed in the introduction from line 50-56 is long, static and inhibits the flow of reading, to improve this section a simple table, naming these factors could be implemented in an appendix.

In line 261-263 the pathophysiological mechanism mentioned is lacking a reference.

Experimental design

In line 73-75 the aim of the review is made. The authors seek out to answer how psychological factors influence AF in terms of progression (meaning severity? What end point?). Is progression well reported in this review – with most emphasis being on causality in my opinion? If not, they should remove it from the aim.

Validity of the findings

In line 276-278, due to AF pathology the mentioned distinction of difference (first onset, paroxysmal and permanent) can be difficult to fulfill because of registration – often only one diagnose code for all cases is possible. Predictions of AF state at onset and further prognosis is problematic. Can the authors elaborate on this or consider at revision?

In line 279-280, the authors mention age as an interesting variable in implementation of future studies. I agree, however where is the authors elaboration on this thought, as it stands alone in the present from as a, more or less, meaningless statement.

Additional comments

The authors Galli et al. attempt to answer if psychological factors is associated with atrial fibrillation (AF) onset, progression, severity and outcomes though a systematic review approach. The paper is well-written and the analysis appropriately justified. This paper provides important information for AF researchers. I only have several minor comments that need to be addressed to accept their review. There are aspects of the text that need to be clarified and possibly changes made to help readability.

Reviewer 2 ·

Basic reporting

No comments.

Experimental design

No comments.

Validity of the findings

No comments.

Additional comments

The idea of the article “to make a systematic review of the numerous studies dealing with psychological factors in AF, in order to identify factors not only linked to AF, but also influencing its onset and progression” is interesting and novel.

For the firs there is an attempt to summarize the existing data considering physiological factors
in AF patients.

The search strategy, selection criteria and data extraction can be accepted.

There is no clear conclusion of the article, the paper rather shows that it is impossible to draw definitive conclusions, which is based on existing data. It’s a massage as well.

“AF is classified as paroxysmal, persistent, and permanent” form Introduction, line 47 - it’s better to include current classification based on 2016 ESC Atrial fibrillation guidelines published in European Heart Journal.

For more clear message the article needs one more language revision.

·

Basic reporting

It has been suggested that psychological factors are associated with the onset and severity of atrial fibrillation, however, there has been no clear evidence regarding that. The authors performed a research of 832 previous articles and found 8 studies for systematic review. The aims of this study were to recognize the involvement of psychological factors and to establish new clinical strategies for the management of atrial fibrillation.
The topic has not been well studied and the hypothesis is well written. However, the introduction, results, and discussion parts are too long, and it is hard to read and understand. Those should be simplified.

Experimental design

The aims of this study were well described in the manuscript and the study was performed with proper methods.

Validity of the findings

Unfortunately the authors did not find any significant risk factors associated with atrial fibrillation because of the heterogeneous methodology for the psychological assessment in the articles they studied. As a consequence, they failed to suggest any clinical intervention to psychological factors which can trigger atrial fibrillation.
A big question in this study was, as the authors suggested, psychological factors such as anxiety or depression were triggering atrial fibrillation or just consequences of atrial fibrillation. That should be addressed further, because clinicians would be interested in intervening any triggering factors that can induce atrial fibrillation, but not as much in treating the consequence of atrial fibrillation.

Additional comments

The topic in this study is very interesting and the authors should be congratulated for their challenging attempt to reveal the association with psychological disorders and atrial fibrillation.
It was unfortunate that the authors did not find any significant risk factors for atrial fibrillation because of the heterogeneous methodology in the previous articles.

---

## Round 0.2 · accepted · Accept

Dear authors,

The article is now Accepted. There was one minor comment from Reviewer 2 which can be resolved while in production.

With respect and warm regards,
Dr Palazón-Bru

Reviewer 2 ·

Basic reporting

The authors took into account all the remarks suggested by the reviewers. All suggested by Peer J points are fulfilled.

Experimental design

No comments.

Validity of the findings

No comments.

Additional comments

As I've previously written, there is no clear conclusion of the article, the paper rather shows that it is impossible to draw definitive conclusions, which is based on existing data.
The AF people are very heterogenous population also due to age and concomitant diseases. Substantial part of them do not feel arrhythmia presence at all. Paradoxically the most serious psychological problems are with paroxysmal and persistent forms of AF. After converting to permanent forms most of the patients feel better and better. They accept the fact that they can live with AF (they realize it) and in longer perspective they do not feel the arrhythmia. Well designed and planned studies are to be done to solve this problem. Quality of life issues are also important.

One minor remark - in the abstract structure "Conclusions" rather than "Discussion" should be used. But better leave it to the Peer J Editor decision.

·

Basic reporting

The draft is still lengthy, but the readability is slightly better than first one.

Experimental design

The study was well designed and well written.

Validity of the findings

No comment.

Additional comments

The authors should be congratulated for a completion of a challenging task. Unfortunately they could not find any psychological risk factor associated with atrial fibrillation due to heterogeneous methodology of previous literature.